# Double-Duty Caregiving, Burnout, Job Satisfaction, and the Sustainability of the Work–Life Balance Among Italian Healthcare Workers: A Descriptive Study

Antonio Urban [1], Mirian Agus [2], Nicola Aru [2], Francesca Corona [2], Elisa Cantone [3], Claudio Giovanni Cortese [4] and Marcello Nonnis [2,*]

1. Cagliari University Hospital, 09124 Cagliari, Italy; a.urban@aoucagliari.it
2. Department of Pedagogy, Psychology, Philosophy, University of Cagliari, 09123 Cagliari, Italy; mirian.agus@unica.it (M.A.); n.aru1@studenti.unica.it (N.A.); f.corona20@studenti.unica.it (F.C.)
3. Department of Medical Sciences and Public Health, University of Cagliari, 09124 Cagliari, Italy; elisa.cantone@libero.it
4. Department of Psychology, University of Turin, 10124 Turin, Italy; claudio.cortese@unito.it
* Correspondence: marcello.nonnis@unica.it; Tel.: +39-070-6757516

**Abstract:** The present study aimed to evaluate the multivariate relationships between variables related to burnout and job stress in healthcare workers, evaluating whether the relationships between these dimensions, the variables related to personal factors (age, seniority of service), and work–family balance factors (overwork related to unused vacation days and accumulated overtime hours) change when the worker is engaged in double-caregiving activities. Indeed, the twofold activities of home caregiving and caring at work might expose workers to challenging situations. To accomplish our aim, we carried out network analyses on data from 466 workers (77.90% females). Participants completed the Link Burnout Questionnaire (LBQ) and the Job Satisfaction Scale (OSI). Contrary to expectations, the variables related to work–life balance played a marginal role with respect to job satisfaction and burnout risk for the whole sample. In addition, no significant differences emerged between workers who reported dual-caregiving tasks compared with those who did not. However, some peculiar aspects of the relationship between burnout and job satisfaction emerged in the two subsamples. The results enable an understanding of the interactions among the assessed variables and allow hypothesizing interventions for the sustainability of the work–life balance in healthcare workers with dual-care tasks.

**Keywords:** sustainable; double-duty caregiving; burnout; fulfilment; job satisfaction; work–life balance; JD-R theory; occupational health; healthcare professionals; network analysis

## 1. Introduction

### 1.1. Double Caregiving and Work–Life Balance in Healthcare

The concept of double caregiving [1,2] refers to how informal caregiving, combined with formal work in healthcare settings, can deteriorate caregivers' mental and physical health, increasing stress, presenteeism, and emotional exhaustion and worsening job satisfaction and performance. Some studies [3–6] suggest that the risk of burnout in caregivers increases when they have to manage multiple roles and work–family conflict is prevalent, potentially increasing detachment and disengaged work-related behaviors instead of actually leaving their jobs (so-called "quiet quitting" [7]). These effects, including the risk of actually quitting the job, also occur in caregivers who benefit in terms of job satisfaction and personal well-being from their work–life balance [5]. Furthermore, Gérain and Zech [6]

found that informal caregivers experience higher levels of burnout than non-caregivers, especially in terms of emotional exhaustion. In particular, women with combined caregiving roles, older caregivers, and those engaged in triple caregiving have reported lower psychosocial well-being than those without family caregiving duties. Double caregivers have also been characterized by higher stress, greater family–work conflict, increased perceived stress and psychological distress, and overall worse psychosocial functioning [8]. Caregivers caring for children (with and without special needs) and individuals with chronic disabilities have shown a particularly high risk of burnout, increased family–work conflict, and a lower relationship quality with partners [6,8]. In healthcare workers, prolonged work–life imbalance contributes to burnout and dissatisfaction with work–life integration [9]. Conversely, a positive work–life balance is a predictor of reduced levels of burnout plus improved mental health and well-being [10–13].

During the COVID-19 pandemic, the phenomenon of burnout among double caregivers was further amplified. Parmar et al. [14] documented how double-shift caregivers, who were already balancing work and informal care, experienced a worsening of their conditions during the pandemic. Emotional and physical overload increased significantly, and most caregivers reported a decline in their mental health and higher levels of anxiety. These results emphasize the need to recognize the crucial contribution of double caregivers and provide them with more support, especially during crises like COVID-19 [14]. Also, the conception of balance—or integration—between private life and work has changed dramatically [15]. In fact, the work–life balance in health professionals was compromised with the recent pandemic, becoming an even more complex goal to define and achieve, especially for these workers who were on the frontlines of the COVID-19 emergency [16,17]. This work context caused healthcare professionals to face several additional challenges, both on socio-relational and personal levels, such as dealing with the fear of contracting the virus or bringing it into their family contexts, the potential stigma, work addiction, and further psychological and physical health issues [16].

Recent studies [18] have identified a positive relationship between work–life balance and job satisfaction while others (involving nurses) have focused on the impact of work–family conflict on life satisfaction [19] or on self-rated health outcomes [20]. Managing work and personal spheres can become even more challenging when one performs a job that requires care but also plays a caregiver role outside the professional field; thus, more attention from organizations is needed [2,14,21]. These ideas, among others, have been recently implemented in the job demands–resources (JD-R) model, wherein the multi-level complexity of occupational well-being is discussed and considered as constantly influenced by the dynamics of different aspects of people's lives and their reciprocal interactions, always in a context of balance [22].

### 1.2. Burnout and Double Caregiving

According to the most widely accepted definition, burnout is a work-related syndrome that manifests through three degenerative aspects: the worker's psychophysical exhaustion, a cynical attitude towards users and colleagues (or depersonalization), and a decline in professional efficacy [23–26]. Over the last two decades, the concept of burnout has evolved and is now recognized as an organizational pathology that affects the entire service sector. This new perspective is based on the JD-R model, which views burnout as the result of an imbalance between job demands (e.g., pressing deadlines or inadequate work environments) and available resources (e.g., decision-making autonomy or perceived organizational support) [24,27].

In the frame of JD-R theory [28,29], job burnout is seen as the antithesis of work engagement, a state characterized by vigor (high mental energy), dedication (attribu-

tion of meaning to work), and absorption (deep concentration in work activities) [30–36]. Recently, the World Health Organization (WHO) included burnout in the ICD-11 as a non-medical condition [25,37], adopting the three dimensions of the model proposed by Maslach et al. [23–26] mentioned above.

Santinello and Negrisolo [38] and Borgogni et al. [39] proposed adding a fourth dimension to the traditional model called disillusion, previously proposed by Edelwich and Brodsky in 1980 [35,40]. This stage of job burnout reflects the erosion of professional ideals and work-related expectations, emphasizing the importance of the meaning that work has for the individual, both socially and existentially. Disillusion is a dimension deeply embedded in the healthcare professions and deserves attention.

Regarding the factors that may influence the syndrome, recent studies focused on healthcare setting have identified, among others, a lack of support within healthcare organizations [41], job duties, skills, treatment received in the workplace, and opportunities for career advancement [42]. Research on burnout among healthcare professionals engaged in double-caregiving activities has highlighted several critical aspects that link work–life conflict and its impact on mental and physical health. Gérain and Zech [4] proposed a theoretical model for understanding informal caregiver burnout, adapting the concept of burnout, usually applied to work environments, to the context of informal caregiving and differentiating informal caregiver burnout from subjective burden, which refers to the subjective perception of care-related stress. The proposed model, which integrates the model of carer stress and burden and the JD-R model, is called the informal caregiving integrative model (ICIM), highlighting the main factors involved in caregiver burnout such as caregiver characteristics, the care setting, and the social environment [4].

Several authors have investigated the effects of the COVID-19 pandemic on the psychological health of healthcare workers. For example, it has been found that burnout rates were higher among nurses during the pandemic and that the major significant predictors were high stress levels and traumatic work experiences [43]. Moreover, Burrowes et al. [44] indicated that 59% of respondents experienced burnout weekly and a substantial number considered leaving the profession within 5 years due to high stress and feeling undervalued. These findings reaffirm the importance of urgently intervening in organizational settings by implementing psychological support systems, mental health interventions for professionals, increased salaries, and flexible schedules [43–45].

### 1.3. Job Satisfaction in Healthcare Professionals

Job satisfaction can be defined as an experience of pleasure related to the accomplishment of something coveted [46]. Along with support from colleagues and work–life balance, job satisfaction is a key dimension of the overall well-being of healthcare professionals [47] and essential to promote so that the quality of healthcare services can also be ensured [47,48].

Several studies have investigated the relationship between job satisfaction and burnout among healthcare workers, suggesting the pivotal and protective role of organizational factors [49–53]. Other studies have found that job satisfaction is closely related to conflict resolution and relationships with colleagues while salary, promotion opportunities, and interpersonal communication have emerged as significant sources of dissatisfaction [54–56].

Regarding the COVID-19 pandemic, a recent study that examined nurses' burnout and job satisfaction revealed that an alarming 91.1% experienced high levels of burnout, significantly impacting their job satisfaction; it has been pointed out that demographic factors and job characteristics are crucial in influencing healthcare workers' levels of burnout and overall job satisfaction [57]. To our knowledge, as argued thus far, there have been few

studies involving the comparison of job satisfaction and the risk of burnout between those who have and those who do not have double-caregiving tasks in a healthcare context.

### 1.4. Study Aim

Because of what has been argued so far, in our view, it is clear that the presence of dual-caregiving tasks can have important implications for the sustainability of healthcare workers' work–life balance, their burnout, and their job satisfaction. With the foregoing literature review, we identified some limitations in the current research on the relationships between burnout and job satisfaction in healthcare providers with dual-caregiving duties compared with those without. Indeed, some previous studies [3–11] mainly followed a confirmatory perspective, identifying independent and dependent variables. This approach neglects the correlational–explorative perspective, ignoring the network structure of the relationships between these variables. Thus, by applying network analysis, our study attempted to fill these gaps, exploring the relationship between burnout and job satisfaction in healthcare professionals with dual-caregiving duties compared with those without.

In addition, from a work–life balance perspective [9,10], the importance of the dimensions 'overwork related to unused vacation days' and 'accumulated overtime hours' emerges, which can describe the interference of work in the private lives of healthcare workers. Furthermore, the variables of age and seniority of service are also of extreme importance with reference to the risk of burnout and job dissatisfaction [49–53], but there have been few studies on the role of these two variables in relation to the dual-caregiving tasks of healthcare workers.

Finally, we wanted to consider age and seniority as separate (though conceptually related) variables given the general aging of the working population [2] and the specific condition of high seniority in the sample (see Section 2.3 below) that characterizes the Italian reality [38], determined jointly by the raising of the retirement age of healthcare workers and the prolonged blockage of their turnover.

Overall, the aims of this study were to describe the multivariate relationships between burnout and job satisfaction dimensions in healthcare workers involved in double caregiving and compare them with those of health workers who did not have these tasks. In particular, we wanted to describe these dimensions in relation to personal (age, seniority of service) and work–family balance (unused vacation days, accumulated overtime hours) variables.

For this reason, we formulated the following research questions regarding the whole sample:

R1—Which nodes (variables) are central to the network?

R2—Which nodes (variables) play the most bridging roles between variables in the network?

R3—Which bridges between variables are strongest?

When comparing the subsample with dual-caregiving tasks versus the subsample without, we came up with the following questions:

R4—Do substantial differences emerge in the centrality of nodes (variables) between the two subsamples?

R5—Are there differences in which nodes (variables) play more of a bridging role between the variables, and between the two subsamples?

R6—Are there differences between the two subsamples in which bridges are stronger between variables?

## 2. Materials and Methods

### 2.1. Research Design

This study was descriptive in nature and was conducted at two public hospital facilities in the province of Cagliari (Sardinia, Italy) as part of a program for the prevention of work-

related stress and burnout risk in healthcare workers. Participants were recruited by a non-probabilistic sampling procedure: training course participants filled out the research protocol voluntarily and according to their availability without receiving any compensation; therefore, no sampling criteria could be followed. Given the specific data collection context, all questionnaires delivered were valid and correctly completed by the participants. The data were collected between September 2023 and July 2024.

### 2.2. Assessment Instruments

The research protocol included two distinct sections. The first was related to the measurement of demographic variables (i.e., age, gender, informal or familiar caregiving activities) and social and professional features (i.e., organizational position, seniority of service, unused vacation days, accumulated overtime hours).

The second section included two assessment instruments standardized and validated in Italy. The Link Burnout Questionnaire (LBQ) [58] was administered to assess the workers' job burnout and work engagement. This questionnaire is a self-assessment of 24 items evaluated with a 6-point scale (from 1 = "Never" to 6 = "Always"). The dimensions evaluated, with each being bipolar and characterized by six items—three positive and three negative—included the following:

-   psychophysical exhaustion–engagement (item examples: "I feel physically exhausted by my work", "Work makes me feel active and vital"; reliability $\alpha = 0.77$);
-   relational deterioration–involvement (item examples: "I have the impression that most of my users do not follow my directions", "I feel gratified by the relationship with my users"; $\alpha = 0.79$);
-   professional inefficacy–efficacy (item examples: "I feel inadequate to deal with my users' problems", "At work, I seem to deal effectively with most of the problems"; $\alpha = 0.78$);
-   disillusion–fulfilment (item examples: "My expectations of this work have been frustrated", "I still feel motivated by my professional ideals"; $\alpha = 0.85$).

Job satisfaction was evaluated with a subscale of the Italian version of the Occupational Stress Indicator (OSI) [59]. This instrument is characterized by five scales:

-   career satisfaction (six items; e.g., "The possibility of maturation or personal development that your job allows you"; reliability $\alpha = 0.77$);
-   satisfaction with the job itself (four items; e.g., "The type of work and the tasks that you are expected to perform"; $\alpha = 0.75$);
-   satisfaction with the setting and the organizational structure (five items; e.g., "The ways in which changes and innovations are implemented"; $\alpha = 0.81$);
-   satisfaction with organizational processes (four items; e.g., "The opportunity to participate in important decisions"; $\alpha = 0.76$);
-   satisfaction with interpersonal relationships (three items; e.g., "Your relationships with others in the work environment"; $\alpha = 0.73$).

The questions were headed by the following sentence: "Rate your level of satisfaction" (evaluated with a Likert scale from 1 = "Extremely unsatisfactory" to 6 = "Extremely satisfactory").

We opted to use the LBQ and OSI for two reasons. The first was the remarkable reliability and robustness of these instruments. In fact, the LBQ and OSI are the best known and most widely used questionnaires in counseling in Italy for the individual assessment of burnout and job satisfaction, respectively. The second reason was the availability of the adaptation, calibration, and validation of these two instruments with a normative sample for the Italian context.

### 2.3. Participants

A total of 466 health professionals took part in the assessment (mean age: 49.57, SD = 9.72; range: 25–67 years). Specifically, 363 (77.90%) participants were female; 103 (22.10%) were male. They reported the following organizational roles: executive physician (*n* = 77, 16.50%), nursing coordinator (*n* = 16, 3.50%), nurse (*n* = 199, 42.70%), obstetrician (*n* = 29, 6.30%), healthcare technician (*n* = 25, 5.40%), and socio-healthcare worker (*n* = 119, 25.60%). They were then divided into different occupational levels: direction (*n* = 12, 2.60%), coordination (*n* = 21, 4.70%), or subordinate (*n* = 432, 92.70%). The participants reported an average value of seniority service of 18.49 (SD = 11.09) years, an average value of unused vacation days of 24.52 (SD = 34.03), and an average value of accumulated overtime hours of 105.16 (SD = 172.73). A total of 202 participants (43.35%) reported that outside of work, they performed a caregiving role, that is, voluntary (unpaid) assistance to relatives (and/or acquaintances) with the following characteristics:

-   with chronic and/or degenerative disabilities of a mental and/or physical nature (e.g., elderly);
-   minors with chronic and/or degenerative disabilities (physical and/or mental) or with special health or educational needs.

Participants with simple tasks of caring for family members in normal circumstances (e.g., family and/or educational caregivers for their own children or self-sufficient parents) were explicitly excluded from this subsample.

### 2.4. Data Analysis

The descriptive features of the variables and scales were inspected (mean, standard deviation, skewness, kurtosis). The descriptive statistics for all assessed variables are reported in detail in Appendix A.

We applied the Spearman's rho coefficient in order to evaluate the bivariate relationships. Furthermore, we computed the Mann–Whitney test in order to explore the potential differences between the medians regarding the groups of workers that were/were not engaged in double-caregiving activities.

The complex relationships among variables were evaluated by the application of network analysis (NA), which can provide valuable insights into the complex interrelationships among psychological variables, individual features, and organizational specificities [60,61]. Indeed, NA allows for the handling of complex, high-dimensional data typical in psychological research, uncovering patterns that traditional statistical methods might miss.

We decided to include, in the NA, some variables related to sociodemographic and professional dimensions (age, seniority of service, unused vacation days, accumulated overtime hours) and the scales involved in the assessments applied (psychophysical exhaustion–engagement, relational deterioration–involvement, professional inefficacy–efficacy, disillusion–fulfilment, career satisfaction, satisfaction with the job itself, satisfaction with the setting and the organizational structure, satisfaction with organizational processes, satisfaction with interpersonal relationships). We chose to include, in the NA, these variables because these are conceptually fundamental for the phenomenon under study.

Furthermore, we decided not to include, in the NA, additional sociodemographic and occupational variables (e.g., gender, department, occupational category). The group of participants did not appear to be balanced and stratified in relation to these variables.

The application of NA can aid in generating new hypotheses about the interplay of psychological factors, guiding future research directions [62]. It might be adequately applied in psychology due to its ability to effectively represent and analyze the complex, multidimensional, and dynamic nature of psychological data. It provides valuable insights that enhance our understanding of psychological and organizational processes, ultimately

contributing to more effective interventions. For these reasons, NA is useful for exploratory data analysis, helping uncover unexpected relationships [63]. NA is designed to capture and analyze these complex relationships, offering insights into how different elements influence each other. NA allows one to illustrate human behavior and psychological dimensions affected by a multitude of factors simultaneously; it can incorporate and analyze this multidimensionality effectively. Furthermore, the effective data representation and the application of graphical models allow one to depict variables as nodes and their interactions as edges (connections), providing a clear and intuitive visualization of complex relationships. This visual representation helps in identifying patterns, clusters, and central elements within psychological data, making complex data more understandable [60,61].

The computation of specific centrality measures allows the identification of key variables; distinctively, we could identify which variables (nodes) were most central or influential in the network. This was crucial to understanding which aspects/variables played pivotal roles in the psychological phenomena in focus. By identifying central nodes, interventions can be more precisely targeted to disturb maladaptive networks and promote positive changes.

Applying NA, each variable of the model is depicted as a node as the connection-relating nodes are illustrated as edges [64]. Conventionally, in the psychological context, blue edges designate positive relationships; red edges imply negative associations. Furthermore, the widths of edges suggest their extents.

In this study, first, we estimated a network that involved the total sample of workers. Next, we applied NA by splitting the sample regarding the variable 'double caregiving', distinguishing the individuals that had reported double-duty caregiving activities and individuals who had not reported caregiving in their families. The analyses were applied with the JASP open-source software (release 0.18.3) [65].

### 2.5. Ethical Issues

This study was approved by the Ethical Committee at Cagliari University, Italy (approval number 0166737 dated 10 July 2023), and was thus conducted in full agreement with the Ethical Principles of Psychologists and the Code of Conduct of the American Psychological Association (APA), joined into the Associazione Italiana di Psicologia (AIP) Code of Ethics. The research was carried out with informed and consenting workers; furthermore, according to Italian law, the project ensured the anonymity and privacy of all contributors.

## 3. Results

### 3.1. Results of the Overall Sample

To evaluate the correlations between variables, the Spearman's rho coefficient was computed, considering the social and professional variables (age, seniority of service, number of unused vacation days, and number of accumulated overtime hours), and the dimensions were assessed with burnout (LBQ) and job satisfaction (OSI) questionnaires (Table 1).

The findings highlighted a significant positive correlation between age and seniority of service (rho = 0.685 ***), and between age and unused vacation days (rho = 0.201 ***); there was a significant negative correlation between age and relational deterioration–involvement (rho = −0.135 ***). Also, seniority of service showed a positive, significant correlation with unused vacation days (rho = 0.194 ***). We also observed a positive, significant correlation between unused vacation days and accumulated overtime hours (rho = 0.355 ***) and a negative correlation between satisfaction with interpersonal relationships and unused vacation days (rho = −0.159 *). We also observed weak significant correlations between unused

vacation days and psychophysical exhaustion–engagement (rho = 0.112 *), professional inefficacy–efficacy (rho = 0.095 *), and career satisfaction (rho = −0.115 *). Furthermore, accumulated overtime hours showed feeble significant correlations with psychophysical exhaustion–engagement (rho = 0.115 *), disillusion–fulfilment (rho = 0.108 *), and satisfaction with the job itself (rho = −0.093 *). Psychophysical exhaustion–engagement correlated positively with all other scales of burnout and correlated negatively with all subscales of job satisfaction (Table 1). The same trend was confirmed for the other scales; specifically, the job satisfaction subscales correlated positively between the instruments and negatively with the burnout dimensions (Table 1).

We analyzed the data to explore the network structure of the variables, starting with the partial correlation matrix. The variables included in the NA were the following: age, seniority of service, unused vacation days, accumulated overtime hours, and the burnout (LBQ) and job satisfaction (OSI) dimensions. In this study, the network structure highlighted variations in relationships between the variables according to the undirected edges (i.e., in which the nodes showed connecting lines, implying some mutual relationships without arrowheads to suggest the direction of influence). The NA was computed by the application of a pairwise Markov random field (PMRF), recognizing nodes that performed as 'ties' between others (i.e., the ties denoted nodes that functioned as single links between two other nodes in this specific network). 'Betweenness' shows the number of shortest paths connecting any two variables.

The quantification of closeness refers to the manner in which a node indirectly is linked to other nodes (i.e., the computation applies the reciprocal of the sum of the smallest pathways from the considered node to different nodes). Closeness is figured as the inverted sum of the total length of all the shortest paths between a particular node and the remaining nodes in the network.

The evaluation of strength aims to identify the nodes that have dense direct links with others (i.e., estimated by the sum of all the absolute edge weights associated with a node). Strength quantifies the sum of the absolute weights of the edge [60,66]. Specifically, a standardized estimation by the extended Bayesian information criterion (EBIC) and the least absolute shrinkage and selection operator (LASSO) [61] were applied [62].

To reveal the importance of each node, centrality indices were considered (i.e., betweenness, closeness, strength, expected influence) [61]. Specifically, the nodes with elevated estimates of centrality indices were judged as the most important nodes in the network.

Betweenness assesses the number of times a node sits on the smallest pathway between two other nodes, relating the node to all the others in the network. The computation of the expected influence is considered to overwhelm the probable fallibility of usual centrality measures in networks with both positive and negative edges [66].

In order to improve the possibility to compare the role of each node, standardized z-scores for all indices were applied [63]. Thus, Zhang's clustering coefficient was computed [67,68] to identify the locally unnecessary nodes in the network. Finally, we considered stability indices and weights by the application of a non-parametric bootstrap procedure using 1000 iterations [61,63]. We considered the accuracy and stability of coefficients, estimating the centrality stability coefficient that should not be below 0.25 and preferably should be above 0.5 [61].

Overall, to better read and interpret the results of NA, tables and graphs must be observed together and integrated to summarize, visualize, and understand the relationships and structure of the network. Indeed, the tables show detailed metrics and attributes, and the graphs allow one to identify visual patterns and relationships to uncover insights like key variables, bridges between groups, group structures, and communication pathways [64].

**Table 1.** Spearman's rho correlations.

| Variable | | 1 | | 2 | | 3 | | 4 | | 5 | | 6 | | 7 | | 8 | | 9 | | 10 | | 11 | | 12 | | 13 |
|---|---|---|---|---|---|---|---|---|---|---|---|---|---|---|---|---|---|---|---|---|---|---|---|---|---|
| 1. Age | Rho | — | | | | | | | | | | | | | | | | | | | | | | | | |
| | *p*-value | — | | | | | | | | | | | | | | | | | | | | | | | | |
| 2. Seniority of service | Rho | 0.685 | *** | — | | | | | | | | | | | | | | | | | | | | | | |
| | *p*-value | <0.001 | | — | | | | | | | | | | | | | | | | | | | | | | |
| 3. Unused vacation days | Rho | 0.201 | *** | 0.194 | *** | — | | | | | | | | | | | | | | | | | | | | |
| | *p*-value | <0.001 | | <0.001 | | — | | | | | | | | | | | | | | | | | | | | |
| 4. Accumulated overtime hours | Rho | −0.051 | | 0.083 | | 0.355 | *** | — | | | | | | | | | | | | | | | | | | |
| | *p*-value | 0.960 | | 0.465 | | <0.001 | | — | | | | | | | | | | | | | | | | | | |
| 5. Psychophysical exhaustion–engagement | Rho | −0.032 | | −0.004 | | 0.112 | * | 0.115 | * | — | | | | | | | | | | | | | | | | |
| | *p*-value | 0.496 | | 0.922 | | 0.016 | | 0.013 | | — | | | | | | | | | | | | | | | | |
| 6. Relational deterioration–involvement | Rho | −0.135 | ** | −0.090 | | 0.012 | | −0.008 | | 0.529 | *** | — | | | | | | | | | | | | | | |
| | *p*-value | <0.004 | | 0.052 | | 0.791 | | 0.872 | | <0.001 | | — | | | | | | | | | | | | | | |
| 7. Professional inefficacy–efficacy | Rho | −0.006 | | −0.050 | | 0.095 | * | −0.002 | | 0.642 | *** | 0.492 | *** | — | | | | | | | | | | | | |
| | *p*-value | 0.889 | | 0.279 | | 0.040 | | 0.972 | | <0.001 | | <0.001 | | — | | | | | | | | | | | | |
| 8. Disillusion–fulfilment | Rho | −0.039 | | 0.029 | | 0.080 | | 0.108 | * | 0.784 | *** | 0.554 | *** | 0.599 | *** | — | | | | | | | | | | |
| | *p*-value | 0.403 | | 0.533 | | 0.086 | | 0.021 | | <0.001 | | <0.001 | | <0.001 | | — | | | | | | | | | | |
| 9. Career satisfaction | Rho | 0.046 | | −0.013 | | −0.115 | * | −0.090 | | −0.657 | *** | −0.447 | *** | −0.482 | *** | −0.730 | *** | — | | | | | | | | |
| | *p*-value | 0.325 | | 0.773 | | 0.013 | | 0.055 | | <0.001 | | <0.001 | | <0.001 | | <0.001 | | — | | | | | | | | |
| 10. Satisfaction with the job itself | Rho | 0.014 | | 0.062 | | −0.072 | | −0.093 | * | −0.675 | *** | −0.486 | *** | −0.550 | *** | −0.673 | *** | 0.725 | *** | — | | | | | | |
| | *p*-value | 0.766 | | 0.184 | | 0.122 | | 0.047 | | <0.001 | | <0.001 | | <0.001 | | <0.001 | | <0.001 | | — | | | | | | |
| 11. Satisfaction with the setting and organizational structure | Rho | 0.056 | | −0.016 | | −0.146 | ** | −0.123 | ** | −0.642 | *** | −0.435 | *** | −0.466 | *** | −0.666 | *** | 0.819 | *** | 0.689 | *** | — | | | | |
| | *p*-value | 0.227 | | 0.723 | | 0.002 | | 0.009 | | <0.001 | | <0.001 | | <0.001 | | <0.001 | | <0.001 | | <0.001 | | — | | | | |
| 12. Satisfaction with organizational processes | Rho | 0.069 | | 0.037 | | −0.095 | * | −0.102 | * | −0.694 | *** | −0.458 | *** | −0.529 | *** | −0.717 | *** | 0.838 | *** | 0.757 | *** | 0.831 | *** | — | |
| | *p*-value | 0.137 | | 0.432 | | 0.042 | | 0.029 | | <0.001 | | <0.001 | | <0.001 | | <0.001 | | <0.001 | | <0.001 | | <0.001 | | — | |
| 13. Satisfaction with interpersonal relationships | Rho | 0.010 | | −0.056 | | −0.159 | *** | −0.106 | * | −0.697 | *** | −0.363 | *** | −0.504 | *** | −0.695 | *** | 0.791 | *** | 0.687 | *** | 0.795 | *** | 0.799 | *** | — |
| | *p*-value | 0.827 | | 0.226 | | <0.001 | | 0.023 | | <0.001 | | <0.001 | | <0.001 | | <0.001 | | <0.001 | | <0.001 | | <0.001 | | <0.001 | | — |

\* $p < 0.05$, \*\* $p < 0.01$, \*\*\* $p < 0.001$.

Regarding the tables, the data about nodes (e.g., centrality measures, betweenness) were useful to identify key variables or clusters of variables. The data about edges in the tables allow one to list connections between nodes with attributes (e.g., weights), which are useful to analyze relationship strength. The indices shown by the metrics and the network properties like centrality, modularity (communities), and density are useful to understand influence, grouping, and connectivity. Regarding the graphs, the nodes represent the variables; their colors and positions can reflect importance, type, or group membership. Furthermore, the edges represent relationships; thickness and color may indicate the strength or type of connection. In this way, the patterns were highlighted, referring to clusters of strongly connected variables; furthermore, the hubs suggest central influencers, and sparse areas show weaker connections [66].

The NA was carried out in different steps [61], initially regarding the total sample, then separately in relation to workers who provided caregiving and those who did not report providing caregiving. The NA computation with our sample highlighted 13 nodes and 37/78 non-zero edges (sparsity: 0.526). In Table 2, we show the z-standardized indices, highlighting the most influential nodes in the network; we computed centrality indices and Zhang's clustering indices.

**Table 2.** Total sample network analysis: centrality and clustering measures per variable, expressed as standardized values (z-scores).

| | Variable | Centrality Measures per Variable | | | | Clustering Measure per Variable |
|---|---|---|---|---|---|---|
| | | Betweenness | Closeness | Strength | Expected Influence | Zhang |
| 1 | Age | 1.641 | −0.679 | −0.008 | 0.753 | −1.475 |
| 2 | Seniority of service | −0.949 | −0.957 | −0.436 | 0.776 | −0.475 |
| 3 | Unused vacation days | 0.408 | −1.605 | −1.527 | −0.634 | −0.879 |
| 4 | Accumulated overtime hours | −0.949 | −2.068 | −2.119 | −1.007 | −1.835 |
| 5 | Psychophysical exhaustion–engagement | −0.085 | 0.696 | 0.889 | −0.355 | −0.281 |
| 6 | Relational deterioration–involvement | 2.011 | 0.981 | −0.669 | −0.981 | 0.107 |
| 7 | Professional inefficacy–efficacy | −0.949 | 0.642 | −0.461 | −0.412 | 0.801 |
| 8 | Disillusion–fulfilment | 0.901 | 1.019 | 0.721 | −0.706 | −0.166 |
| 9 | Career satisfaction | −0.209 | 0.526 | 0.985 | 1.180 | 0.898 |
| 10 | Satisfaction with the job itself | −0.332 | 0.486 | 0.389 | −1.370 | 0.075 |
| 11 | Satisfaction with the setting and organizational structure | −0.949 | 0.204 | 0.519 | 1.638 | 1.593 |
| 12 | Satisfaction with organizational processes | −0.702 | 0.247 | 1.135 | 1.294 | 0.874 |
| 13 | Satisfaction with interpersonal relationships | 0.161 | 0.507 | 0.583 | −0.178 | 0.763 |

The graphical representation of these relationships is shown in Figure 1; the blue edges and red edges define the positive and negative multivariate partialized relationships among variables, respectively. The stability of estimated centrality indices was assessed and is reported in Figure 2. Supplementary graphical outputs are reported in the Appendix A.

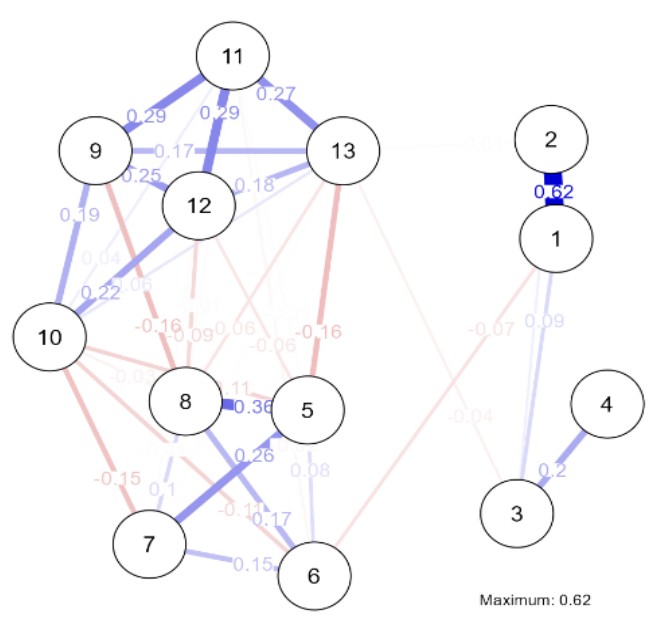

1  Age

2  Seniority of service

3  Unused vacation days

4  Accumulated overtime hours

5  Psychophysical exhaustion–engagement

6  Relational deterioration–involvement

7  Professional inefficacy–efficacy

8  Disillusion–fulfilment

9  Career satisfaction

10 Satisfaction with the job itself

11 Satisfaction with the setting and organizational structure

12 Satisfaction with organizational processes

13 Satisfaction with interpersonal relationships

**Figure 1.** Estimated network model with the total sample. The light blue color of the links between nodes indicates a positive relationship, the red color a negative relationship.

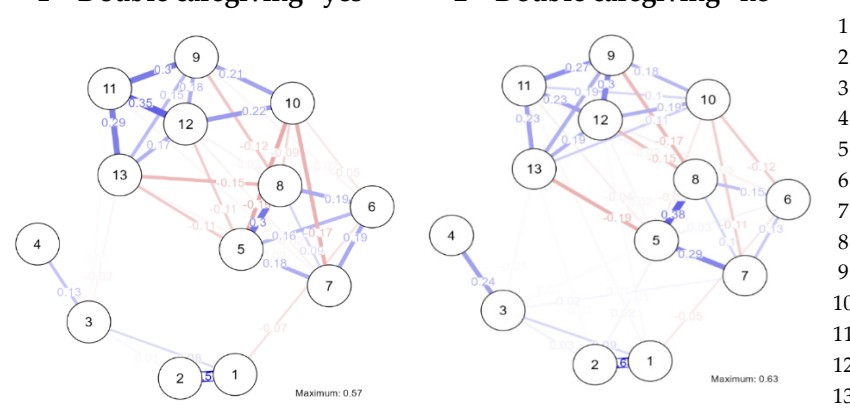

**1—Double caregiving "yes"**      **2—Double caregiving "no"**

1  Age
2  Seniority of service
3  Unused vacation days
4  Accumulated overtime hours
5  Psychophysical exhaustion–engagement
6  Relational deterioration–involvement
7  Professional inefficacy–efficacy
8  Disillusion–fulfilment
9  Career satisfaction
10 Satisfaction with the job itself
11 Satisfaction with the setting and organizational structure
12 Satisfaction with organizational processes
13 Satisfaction with interpersonal relationships

**Figure 2.** Estimated network plots for the two groups. The light blue color of the links between nodes indicates a positive relationship, the red color a negative relationship.

The higher edge is represented by the positive partial coefficient between age and seniority of service, which are moreover positioned in a marginal position in the network. Also, we noted that the variables related to the overwork conditions (node 3—unused vacation days and node 4—accumulated overtime hours) are placed in a marginal position in the network. The central position might be identified for node 8 (disillusion–fulfilment).

Node 8 (disillusion–fulfilment) and node 10 (satisfaction with the job itself) might be considered the "bridges" among burnout and job satisfaction dimensions. The stronger positive index of strength is shown by node 12 (satisfaction with organizational processes). The strong values of expected influences are observed in relation to node 11 (satisfaction with the setting and the organizational structure), node 12 (satisfaction with organizational processes), and node 9 (career satisfaction). The node with high redundance in the network is number 11 (satisfaction with the setting and the organizational structure).

Based on the values shown in Table 2 with reference to the total sample and in particular to the betweenness centrality measure, it can be seen that the relational deterioration–involvement (node 6) and age (node 1) were the variables with the highest values and were therefore central in relation to the entire network. Regarding closeness, it can be observed that the data on unused vacation days (node 3), accumulated overtime hours (node 4), and seniority (node 2) were also quite inflated per standardized negative values—and were less central nodes that had a limited impact on the network—while the opposite could be said for the index of the disillusion–fulfilment dimension (node 8).

Regarding strength, unused vacation days and accumulated overtime hours remain marginal nodes in the network; they appear to have few direct links with the other nodes. However, satisfaction with organizational processes (node 12) appears to have dense direct links with the other nodes.

Regarding expected influence (i.e., how much a node is able to influence the others in both a direct and indirect sense), a small influence of the variables 'accumulated overtime hours' (node 4) and 'satisfaction with the job itself' (node 10) can be observed. Conversely, the highest influence can be seen for career satisfaction, satisfaction with the organizational setting and structure, and satisfaction with the organizational processes (nodes 9, 11, and 12, respectively). Going further into the relationships between the individual nodes of the evaluated psychological dimensions (Figure 1), we noted the inverse relationship between node 5 (psychophysical exhaustion–engagement) and node 13 (satisfaction with interpersonal relationships), the inverse relationship between node 9 (career satisfaction) and node 8 (disillusion–fulfilment), and the inverse relationship between node 7 (professional inefficacy–efficacy) and node 10 (satisfaction with the job itself).

### 3.2. Results of the Subsample Comparison

In order to explore and deepen the potential features that might characterize workers that did/did not engage in double-caregiving activities, we chose to compare the medians of these two groups by the application of a Mann–Whitney test regarding the variables considered in our NA. Table 3 shows that there was a significant difference only in relation to the variables 'seniority of service' and 'unused vacation days', in which the values were higher for workers that reported double-duty caregiving activities. For all the other variables assessed, we did not observe any significant difference between the two groups.

Then, NA was applied, distinguishing the variable double caregiving as "yes" or "no". Specifically, we set the double-caregiving activity as a 'yes/no' splitting variable; we applied this new analysis by using the same statistical setting mentioned previously to try to reveal the multivariate relationships among the variables in the two groups of workers.

For group 1 (double caregiving "yes"), we obtained 13 nodes, with 34 non-zero edges out of 78 (sparsity: 0.564); for group 2 (double caregiving "no"), we observed 13 nodes, with 38 non-zero edges out of 78 (sparsity: 0.513).

Table 4 illustrates the standardized centrality and clustering measures per variable in relation to each group. The graphical representations of the two networks are reported in Figure 2.

Consistently, with the results obtained by the application of the Mann–Whitney test comparison between the two groups (see Table 3), the networks of the two groups appeared similar. To carry out a network comparison, we considered the weight matrix of both networks, and we applied Pearson's linear correlations on them as a measure of similarity [69,70]. The obtained coefficients ranged from 0.840 to 1.000, highlighting the similarity between the two networks.

**Table 3.** Mann–Whitney test comparison between two groups (workers who did/did not provide caregiving).

| | Variables | Statistic | df | *p* | Effect Size (Rank Biserial Correlation) | Group | Mean | Standard Deviation |
|---|---|---|---|---|---|---|---|---|
| 1 | Age | 27,286,500 | 464 | 0.666 | 0.023 | 1—Double caregiving yes | 50.158 | 8.783 |
| | | | | | | 2—Double caregiving no | 49.133 | 10.388 |
| 2 | Seniority of service | 30,544,000 | 464 | 0.007 ** | 0.146 | 1—Double caregiving yes | 20.015 | 10.428 |
| | | | | | | 2—Double caregiving no | 17.319 | 11.454 |
| 3 | Unused vacation days | 30,335,500 | 463 | 0.008 ** | 0.142 | 1—Double caregiving yes | 27.713 | 35.252 |
| | | | | | | 2—Double caregiving no | 22.087 | 32.914 |
| 4 | Accumulated overtime hours | 24,477,500 | 456 | 0.352 | −0.050 | 1—Double caregiving yes | 100.111 | 156.760 |
| | | | | | | 2—Double caregiving no | 109.039 | 184.263 |
| 5 | Psychophysical exhaustion–engagement | 25,144,000 | 464 | 0.291 | −0.057 | 1—Double caregiving yes | 19.421 | 6.189 |
| | | | | | | 2—Double caregiving no | 19.909 | 6.199 |
| 6 | Relational deterioration–involvement | 27,855,500 | 464 | 0.406 | 0.045 | 1—Double caregiving yes | 16.158 | 3.705 |
| | | | | | | 2—Double caregiving no | 15.758 | 3.657 |
| 7 | Professional inefficacy–efficacy | 25,134,000 | 464 | 0.288 | −0.057 | 1—Double caregiving yes | 14.847 | 5.293 |
| | | | | | | 2—Double caregiving no | 15.330 | 5.178 |
| 8 | Disillusion–fulfilment | 26,258,000 | 464 | 0.778 | −0.015 | 1—Double caregiving yes | 14.094 | 4.285 |
| | | | | | | 2—Double caregiving no | 14.057 | 4.345 |
| 9 | Career satisfaction | 25,225,500 | 464 | 0.315 | −0.054 | 1—Double caregiving yes | 10.554 | 2.782 |
| | | | | | | 2—Double caregiving no | 10.795 | 2.779 |
| 10 | Satisfaction with the job itself | 26,930,000 | 464 | 0.854 | 0.010 | 1—Double caregiving yes | 20.411 | 7.628 |
| | | | | | | 2—Double caregiving no | 20.208 | 7.317 |
| 11 | Satisfaction with the setting and organizational structure | 25,526,500 | 464 | 0.429 | −0.043 | 1—Double caregiving yes | 15.916 | 5.685 |
| | | | | | | 2—Double caregiving no | 16.125 | 5.408 |
| 12 | Satisfaction with organizational processes | 23,870,500 | 464 | 0.052 | −0.105 | 1—Double caregiving yes | 13.302 | 4.897 |
| | | | | | | 2—Double caregiving no | 14.125 | 5.177 |
| 13 | Satisfaction with interpersonal relationships | 26,507,500 | 464 | 0.914 | −0.006 | 1—Double caregiving yes | 17.569 | 8.142 |
| | | | | | | 2—Double caregiving no | 17.568 | 7.669 |

** *p* < 0.01.

**Table 4.** Centrality and clustering measures per variable, expressed as standardized z-score values in two groups (1—double caregiving "yes"; 2—double caregiving "no").

| | | 1—Double Caregiving "Yes" | | | | | 2—Double Caregiving "No" | | | | |
|---|---|---|---|---|---|---|---|---|---|---|---|
| | | Centrality | | | | Clustering | Centrality | | | | Clustering |
| | Variable | Betweenness | Closeness | Strength | Expected Influence | Zhang | Betweenness | Closeness | Strength | Expected Influence | Zhang |
| 1 | Age | 1.853 | −0.484 | −0.165 | 0.646 | −1.705 | 1.572 | −0.752 | 0.022 | 0.910 | −1.280 |
| 2 | Seniority of service | −0.865 | −0.743 | −0.610 | 0.646 | −0.313 | −0.761 | −0.955 | −0.357 | 0.859 | −0.742 |
| 3 | Unused vacation days | 0.166 | −1.623 | −1.720 | −0.717 | −1.244 | 0.124 | −1.643 | −1.431 | −0.495 | −0.851 |
| 4 | Accumulated overtime hours | −0.865 | −2.097 | −2.123 | −0.923 | −1.820 | −0.761 | −1.927 | −2.015 | −1.130 | −1.577 |
| 5 | Psychophysical exhaustion–engagement | −0.584 | 0.977 | 0.963 | −0.550 | 0.128 | 0.124 | 0.762 | 0.778 | −0.320 | −0.528 |
| 6 | Relational deterioration–involvement | 2.134 | 1.051 | −0.283 | 0.025 | 0.636 | 1.814 | 0.996 | −0.994 | −1.721 | −0.327 |
| 7 | Professional inefficacy–efficacy | 0.072 | 0.754 | −0.228 | −0.580 | 0.744 | −0.681 | 0.602 | −0.473 | −0.260 | 0.525 |
| 8 | Disillusion–fulfilment | 0.541 | 1.039 | 0.681 | −0.705 | 0.108 | 1.572 | 1.136 | 0.808 | −0.802 | −0.353 |
| 9 | Career satisfaction | −0.865 | 0.255 | 0.710 | 1.013 | 0.771 | −0.118 | 0.530 | 1.183 | 1.278 | 1.063 |
| 10 | Satisfaction with the job itself | −0.022 | 0.577 | 0.556 | −1.620 | −0.002 | −0.761 | 0.594 | 0.292 | −0.838 | 0.500 |
| 11 | Satisfaction with the setting and organizational structure | −0.865 | 0.003 | 0.661 | 1.780 | 1.176 | −0.761 | −0.035 | 0.553 | 1.200 | 1.524 |
| 12 | Satisfaction with organizational processes | −0.584 | 0.015 | 1.048 | 1.317 | 0.674 | −0.761 | 0.417 | 1.107 | 1.156 | 1.113 |
| 13 | Satisfaction with interpersonal relationships | −0.115 | 0.277 | 0.511 | −0.331 | 0.848 | −0.600 | 0.274 | 0.527 | 0.163 | 0.934 |

As previously emphasized, the variables related to age, seniority of service (nodes 1 and 2), and overwork (node 3—unused vacation days; node 4—accumulated overtime hours) were in marginal positions in the network. Moreover, the central position in both networks was held by node 5 (psychophysical exhaustion–engagement) and node 8 (disillusion–fulfilment).

The Zhang's clustering coefficient (for the total sample of workers) for individuals involved or not involved in double-duty caregiving converged in designating node 11 (satisfaction with the setting and organizational structure) as having higher redundance in the network, possibly since the other nodes, its neighbors (9—career satisfaction; 12—satisfaction with organizational processes; and 13—satisfaction with interpersonal relationships), tended to be powerfully associated with each other [67]. In group 2 (double caregiving "no"), the positive associations between nodes 5 (psychophysical exhaustion–engagement) and 8 (disillusion–fulfilment), between nodes 5 (psychophysical exhaustion–engagement) and 7 (professional inefficacy–efficacy), and between nodes 9 (career satisfaction) and 12 (satisfaction with organizational processes) were stronger than in group 1.

Observing in detail the two networks in Figure 2, we highlight, in the group of workers that reported double-caregiving activities (group 1), that there was a strong negative relationship between nodes 5 (psychophysical exhaustion–engagement) and 10 (satisfaction with the job itself). Furthermore, in group 1, we found a strong negative association between nodes 8 (disillusion–fulfilment) and 13 (satisfaction with interpersonal relationships) that was not present in group 2.

In group 1, the positive association between nodes 11 (satisfaction with the setting and organizational structure) and 12 (satisfaction with organizational processes) was stronger than in group 2. Also in group 1, the negative association between nodes 5 (psychophysical exhaustion–engagement) and 13 (satisfaction with interpersonal relationships) was weaker than the same association in group 2.

In group 1, the negative association between nodes 8 (disillusion–fulfilment) and 12 (satisfaction with the organizational processes) was less intense than the association in group 2. Also in group 1, the negative association between nodes 7 (professional inefficacy–efficacy) and 10 (satisfaction with the job itself) appeared stronger than the association in group 2. Finally, in group 1, the negative association between nodes 6 (relational deterioration–involvement) and 10 (satisfaction with the job itself) was weaker than the association in group 2.

## 4. Discussion

### 4.1. Overall Sample

Our findings on the relationship between job satisfaction and job burnout dimensions can provide valuable insights regarding occupational and organizational health in healthcare contexts. Contrary to what we expected and what was found in other studies [3–6], the stressor variables regarding the work–family balance (especially unused vacation days and accumulated overtime hours) played a marginal role in influencing the relationships between the nodes of both burnout and job satisfaction dimensions. The explanation of this (i.e., counter-intuitive) result can be facilitated by the subsample comparison analysis (double caregiving yes/no) presented below (Section 4.2).

The relationship between disillusion–fulfilment and satisfaction with the job itself suggests the importance of the preservation of vocational ideals within the context of healthcare roles and the recognition of health workers' expectations towards their work [52,53]. Also, the centrality of the disillusion–fulfilment dimension emerged, especially its relationship with career satisfaction, like in recent research on the subject [71], satisfaction with

organizational processes, and interpersonal relationships. These findings emphasize the importance of cultivating motivational and vocational aspects of the health professions besides the importance of relationships with colleagues and users/patients, which deserve further research attention.

The inverse relationship between psychophysical exhaustion–engagement and satisfaction with interpersonal relationships reminded us of the value of the emotional dimension in workplace relationships for occupational well-being, satisfaction, and other organizational nature outcomes [72–76]. Finally, the relationship between professional inefficacy–efficacy and satisfaction with the job itself pertains to being able to effectively read the specific problems of one's own context and feeling competent, as well as translating into a better service what to do with the perception of one's own job as a source of satisfaction. In the context of the JD-R model [22], perceived professional efficacy can be considered a resource and, in this sense, a motivating factor, in turn promoting involvement and well-being, mitigating the burden of excessive job demands through a better management of these. Moreover, in the frame of self-determination theory [77,78], regarding the need for competence, a worker who feels a high level of success and mastery in their job experiences greater intrinsic motivation and therefore greater job satisfaction.

### 4.2. Comparison of Subsamples

The results of our NA showed no significant differences in terms of risk of burnout and job satisfaction between the group of healthcare professionals with a double-caregiving role and those without. This result was partially mirrored, for example, in a study by Boumans and Dorant [1], where no significant differences were found in job satisfaction and motivation among healthcare professionals with double-caregiving roles; still, the latter were found to experience greater emotional exhaustion and lower psychophysical well-being. This result was also found in other qualitative studies [2]. However, in the groups we compared, the dimensions of disillusion–fulfilment and psychophysical exhaustion–engagement played a central role. This finding gains meaning in relation to the caring profession that our participants shared.

An interesting finding, observable in the network and arrangement of variables, was the marginal position of the sociodemographic variables, which did not show a significant weight in the relationship with the other variables investigated. The common redundancy of the dimension of satisfaction with the organizational setting and structure could highlight the importance attached by our participants to the workplace and these specific aspects, which stand as predictors of distress [79].

It is also interesting to note that for the healthcare professionals who did not have dual-care duties, the positive relationships between psychophysical exhaustion–engagement and disillusion–fulfilment, between psychophysical exhaustion–engagement and professional inefficacy–efficacy, and between career satisfaction and satisfaction with organizational processes were more intense compared to those for the non-dual-carers. Perhaps the non-caregivers were more vulnerable as their work could be the main source from which they derived well-being and on which motivation and perceived efficacy were modulated, increasing, for them, the risk of experiencing psychophysical exhaustion. Instead, double caregiving could be a protective factor against the more damaging effects of burnout on vocational ideals and perceived professional efficacy as it could allow for the development of greater capabilities and resilience, which are personal resources [79]. The double-caregiving role could grant the opportunity to find meaning outside of the work sphere, therefore not constituting an "additional burden" for the double caregiver. Playing an informal caregiving role could also paradoxically constitute a kind of detachment from the frustrations related to the professional sphere whereas non-double caregivers may focus mainly on the

professional domain and experience greater identification with the formal caregiving role and struggle more in dealing with stressors despite identification with one's work being considered a characteristic of engagement [79]. In the context of JD-R theory, in fact, this result might be argued for by considering that people have specific personal and professional resources and can interpret a greater workload or personal demands as challenges and feel more motivated or turn them into resources. Perhaps a double-caregiving role can both protect healthcare professionals from the risk of burnout and also exacerbate it. In any case, this result, which is at odds with previous studies on dual carers [3–6], requires further research.

Similar reasoning might be applied to the dimensions related to satisfaction. Caregivers who are not dual carers and who have a greater focus on their work environments and their stability might derive more satisfaction from an organization that functions efficiently and thus supports their professional development. Double caregivers, on the other hand, who have to manage a double role, might consider other aspects more important than career opportunities, such as work flexibility and organizational support, thus considering aspects strictly related to organizational processes less crucial to their well-being and work–life balance. These findings also require further research.

The same aspects can be discussed by looking more specifically at the differences that emerged between the two groups (e.g., the negative relationship between disillusion–fulfilment and satisfaction with interpersonal relationships) that subsisted only for double caregivers. This result highlights the relevance of the socio-relational aspect in nurturing and protecting the vocational aspects of these workers, who, being busy juggling professional and informal responsibilities, might find the quality of relationships between coworkers and users more influential on their occupational well-being. However, despite the fact that in general—for both groups—positive relationships at work were associated with less psychophysical exhaustion–engagement, for those with a double-caring role, the protective function of the relational factor was less impactful on this specific dimension of burnout. Similar aspects have emerged in qualitative analyses like that by Detaille et al. [2]. Moreover, relational deterioration–involvement seemed to have less of an impact on satisfaction with the job itself in the group of double caregivers possibly due to, as argued earlier, greater resilience or a greater entrenchment of the satisfaction that can be drawn from the caring role in general or the presence of more developed coping strategies for dealing with relational stress.

Also, for the double caregivers, feeling ineffective had a more pronounced impact on satisfaction with the job itself than the other group of colleagues. This may have been due to some kind of conflict between their formal and informal roles, which, on the other hand, may not have subsisted in those who did not perform the double-caring role since the perception of efficacy could be limited to the work context, thus having less impact on this type of satisfaction. Also, the double caregivers seemed to be less sensitive to the effects of experienced disillusion–fulfilment on satisfaction with organizational processes, probably finding their purpose in other aspects of their existence or professional contexts, as previously discussed. These results also need additional investigation.

Finally, when healthcare professionals reported they were satisfied with their organizational settings and structures, they also tended to be satisfied with the processes taking place, although this effect emerged more significantly for those with the double-caring role, and a functional organization could allow these people to better address and balance it. This finding is in line with JD-R theory in relation to job design [79].

### 4.3. Practical Implications

The results of this study allowed us to outline some possible lines of intervention for the promotion of job satisfaction and the prevention of burnout risk in health workers, which can be considered (albeit with different nuances) as valid for those with dual-caring duties as for those without this role. Regarding the dimension of disillusion–fulfilment, to date little-studied [71], and its relationship with the different dimensions of job satisfaction, it becomes evident how important it is to promote organizational actions that nurture the sense of importance and vocational work motivation in care workers regardless of whether they have double-caregiving tasks.

Regarding psychophysical exhaustion–engagement, some studies have proposed training interventions aimed at enhancing emotional intelligence among health professionals [72]. Through such interventions, healthcare workers could improve their ability to recognize and manage emotions in crisis situations [73]. In addition, emotional competencies could enhance the ability to relate in work teams, particularly regarding the use of empathy and active listening, perhaps to support a colleague who is struggling with work–life balance or on the relational level with patients [74,75]. These skills could also help healthcare workers express their emotional states clearly, transparently, and assertively, fostering a protective mechanism against emotional exhaustion and relational deterioration [76]. In summary, these interventions could be a valuable strategy to ensure these health workers' health is protected, reduce burnout levels, and improve the quality of care of services and relationships with patients and colleagues.

Moreover, in healthcare organizations, satisfaction with the setting and the organizational structure is a central aspect of occupational well-being. Work environments and occupational roles should be individual-appropriate to avoid high levels of distress and inspire the greater involvement of staff and leaders to define ergonomic and structural aspects. Such collaboration can be a protective factor and allow for more efficient adjustments and the better adaptation of planned activities. They can help in planning interventions and/or discussing issues together around how innovations and changes within a company are introduced, following an approach of greater involvement of the staff, implicitly knowing which process improvements can take place and how [43–45].

Concerning the relationship between professional inefficacy–efficacy and satisfaction with the job itself, we recommend promoting a greater perception of professional effectiveness through interventions aimed at improving self-efficacy, autonomy, psychological capital, and self-determination. These skills could help healthcare professionals in the construction of career paths and personalization of their tasks, aspects that could be fostered by incentivizing coordinating figures to exercise a less autocratic leadership more inclined to decentralize power and delegate [49]. Finally, it is certainly important to consider the specific and particular conditions of health workers with more burdensome family circumstances (e.g., those with dual-caregiving duties) and to promote organizational and work actions that enable these people to manage their work–life balance in a sustainable way. For instance, an organization could guarantee more flexibility in working hours, operate strategically and synergistically with workers with respect to vacations, limit overtime hours if considered excessive, and provide psychological support in the working context [45,50]. It is worth emphasizing that any intervention should involve policymakers, who could allocate more resources to facilitate a better management of the health sector with funding that is adequate to the context and in line with the real needs of health workers and citizens, commit to fight precariousness, ensure a periodical monitoring system of occupational well-being such as via work-related stress assessment, and promote occupational health [47,51], providing for a widespread presence of professional figures in the psychological field within health organizations and nationwide.

*4.4. Study Limitations*

This study had several limitations, including the study's mainly descriptive nature, given the still under-explored phenomenon in relation to the specific dimensions assessed, the profession of the participants, and the specific cultural context; thus, we consider our conducted analyses preliminary. Regarding the characteristics of the sample, for a more accurate reading of the results, it is worth noting that most of the participants reported nursing as their job and identified with the female gender, although these variables were not specifically included in the network; similarly, more than 90% of our participants reported no coordinating role, so subsequent research could focus more on health directors or managers, including those who have administrative roles in hospital facilities.

Future research on work–life balance, as well as on the specific topic of double-duty caregiving, could also take into account the variables excluded in this study (the reasons are given in Section 2.4), such as gender and occupational category, or other more specific categories, such as any restorative activity carried out during extra-work hours, that could deepen this area of research. In future studies, it will also be necessary to investigate the different possible declinations of double-duty caregiving. In fact, it is plausible that the caregiving load for the caregiver may be different depending on the type of health problem of the person cared for outside the workplace. In addition, off-the-job caregiving tasks can also add up (and this, as argued, can be configured as triple-caregiving).

Another aspect for further investigation that could not be evaluated here as a single 'macro-factor' could be some kind of strain variable, which might be analyzed as a composite variable, consisting of similar data—or additionally—to those collected in this study, like unused vacation days, accumulated overtime hours, and seniority of service. Moreover, adding a specific measure of double-duty caregiving and work–life balance—or, better, work–life integration—through dedicated instruments, here missing, could lead to more accurate interpretations. Finally, regarding the relationships discussed above, in addition to understanding whether they are replicable in similar studies involving healthcare professionals, they clearly need to be further explored also in qualitative terms to better investigate their nature.

## 5. Conclusions

Given our findings and the theoretical framework of the present study, with reference to work–life balance and JD-R theory, some concluding remarks can be made. Irrespective of whether one has a dual-caring role or not, an organization that is functionally structured such that process design involves more of the employees could constitute a context capable of promoting and facilitating the development of personal and job resources aimed at better balancing private life and work. In public healthcare, greater care for ergonomics and settings could significantly improve the occupational well-being and job satisfaction of healthcare workers while at the same time enabling the better management of stressors, resulting in more caring and respectful environments for workers' needs, expectations, and values. Regardless of the domain from which the demands come, a more sustainable organization of work can in fact be helpful in mitigating the perceived conflict between life and work, and vice versa, reducing the risk of burnout and promoting professional fulfilment, as well as higher levels of engagement and, given the particular times and the current Italian socioeconomic context, lower levels of turnover. Finally, the role of dual caregiving in managing the work–family interface appears complex and multifaceted. In fact, it should not simply be considered an additional demand for this type of healthcare worker, but perhaps has nuances of a vocational nature that can be linked to the self-actualizing aspects of the different healthcare professions. Indeed, it might paradoxically constitute a personal resource that can modulate the relationship between the burnout risk

and job satisfaction of healthcare workers. For this reason, it should be properly considered in the organization and management of the sustainability of the work–family interface of healthcare workers.

**Author Contributions:** Conceptualization, A.U. and M.N.; Methodology, M.A., N.A. and F.C.; Validation, C.G.C. and M.N.; Formal analysis, M.A., N.A. and F.C.; Investigation, N.A., F.C. and M.N.; Resources, A.U. and C.G.C.; Data curation, N.A. and F.C.; Writing—original draft, M.A., N.A., F.C. and M.N.; Writing—review & editing, E.C. and C.G.C.; Supervision, A.U. and C.G.C. All authors have read and agreed to the published version of the manuscript.

**Funding:** This research received no external funding.

**Institutional Review Board Statement:** The study was conducted in accordance with the Declaration of Helsinki and approved by the Ethics Committee of the University of Cagliari (approval number 0166737 dated 10 July 2023) for studies involving humans.

**Informed Consent Statement:** Informed consent was obtained from all subjects involved in the study.

**Data Availability Statement:** Data are available within the article. For more information, please contact the corresponding author.

**Conflicts of Interest:** The authors declare no conflicts of interest.

# Appendix A

**Table A1.** Descriptive statistics.

| | Valid | Missing | Mean | 95% Confidence Interval Mean | | Std. Deviation | Skewness | Std. Error of Skewness | Kurtosis | Std. Error of Kurtosis | Minimum | Maximum |
| | | | | Upper | Lower | | | | | | | |
|---|---|---|---|---|---|---|---|---|---|---|---|---|
| Age | 466 | 0 | 49.577 | 50.487 | 48.695 | 9.728 | −0.597 | 0.113 | −0.508 | 0.226 | 25.000 | 67.000 |
| Seniority of Service | 466 | 0 | 18.488 | 19.470 | 17.490 | 11.090 | 0.142 | 0.113 | −1.100 | 0.226 | 0.000 | 40.000 |
| Unused vacation days | 465 | 1 | 24.531 | 27.611 | 21.628 | 34.027 | 3.164 | 0.113 | 13.739 | 0.226 | 0.000 | 250.000 |
| Accumulated overtime hours | 458 | 8 | 105.159 | 121.974 | 89.474 | 172.727 | 4.585 | 0.114 | 35.718 | 0.228 | 0.000 | 2000.000 |
| OSISCTot Career Satisfaction | 466 | 0 | 19.697 | 20.247 | 19.150 | 6.193 | 0.093 | 0.113 | −0.316 | 0.226 | 6.000 | 36.000 |
| OSISJTot Satisfaction with the Job itself | 466 | 0 | 15.931 | 16.275 | 15.590 | 3.679 | −0.377 | 0.113 | 0.509 | 0.226 | 4.000 | 24.000 |
| OSISSTot Satisfaction with the setting and the organizational Structure | 466 | 0 | 15.120 | 15.612 | 14.648 | 5.228 | 0.023 | 0.113 | −0.429 | 0.226 | 5.000 | 30.000 |
| OSISPTot Satisfaction with organizational Processes | 466 | 0 | 14.073 | 14.451 | 13.684 | 4.315 | −0.017 | 0.113 | −0.452 | 0.226 | 4.000 | 24.000 |
| OSISRTot Satisfaction with interpersonal Relationships | 466 | 0 | 10.691 | 10.966 | 10.455 | 2.780 | −0.066 | 0.113 | −0.060 | 0.226 | 3.000 | 18.000 |
| LBQEPtot Psychophysical exhaustion-engagement | 466 | 0 | 20.296 | 20.940 | 19.644 | 7.446 | 0.055 | 0.113 | −0.952 | 0.226 | 6.000 | 36.000 |
| LBQDRtot Relational deterioration-involvement | 466 | 0 | 16.034 | 16.520 | 15.566 | 5.525 | 0.503 | 0.113 | −0.038 | 0.226 | 6.000 | 34.000 |
| LBQIPtot Professional inefficacy-efficacy | 466 | 0 | 13.768 | 14.255 | 13.307 | 5.069 | 0.972 | 0.113 | 1.309 | 0.226 | 6.000 | 35.000 |
| LBQDStot Disillusion-fulfilment | 466 | 0 | 17.569 | 18.281 | 16.854 | 7.869 | 0.312 | 0.113 | −1.001 | 0.226 | 6.000 | 36.000 |

LBQEPtot = psychophysical exhaustion–engagement; LBQDRtot = relational deterioration–involvement; LBQIPtot = professional inefficacy–efficacy; LBQDStot = disillusion–fulfilment; OSISCtot = career satisfaction; OSISJtot = satisfaction with the job itself; OSISStot = satisfaction with the setting and the organizational structure; OSISPtot = satisfaction with organizational processes; OSISRtot = satisfaction with interpersonal relationships.

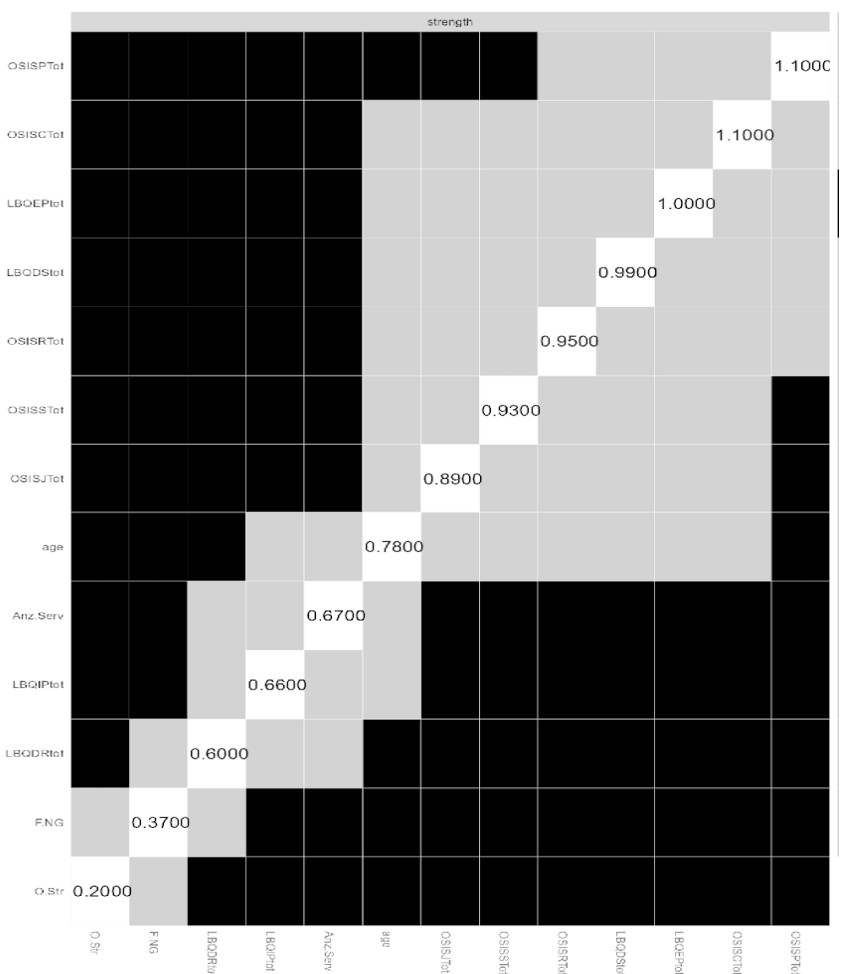

**Figure A1.** Centrality stability coefficients per variable in the network obtained by non-parametric bootstrapped difference test for strength (gray squares designate no difference between nodes; black squares identify significant difference for $\alpha = 0.05$; in the diagonal, we observe the strength values of each node). Anz.Serv = seniority of service; O.Str = accumulated overtime hours; F.NG = unused vacation days; LBQEPtot = psychophysical exhaustion–engagement; LBQDRtot = relational deterioration–involvement; LBQIPtot = professional inefficacy–efficacy; LBQDStot = disillusion–fulfilment; OSISCtot = career satisfaction; OSISJtot = satisfaction with the job itself; OSISStot = satisfaction with the setting and the organizational structure; OSISPtot = satisfaction with organizational processes; OSISRtot = satisfaction with interpersonal relationships.

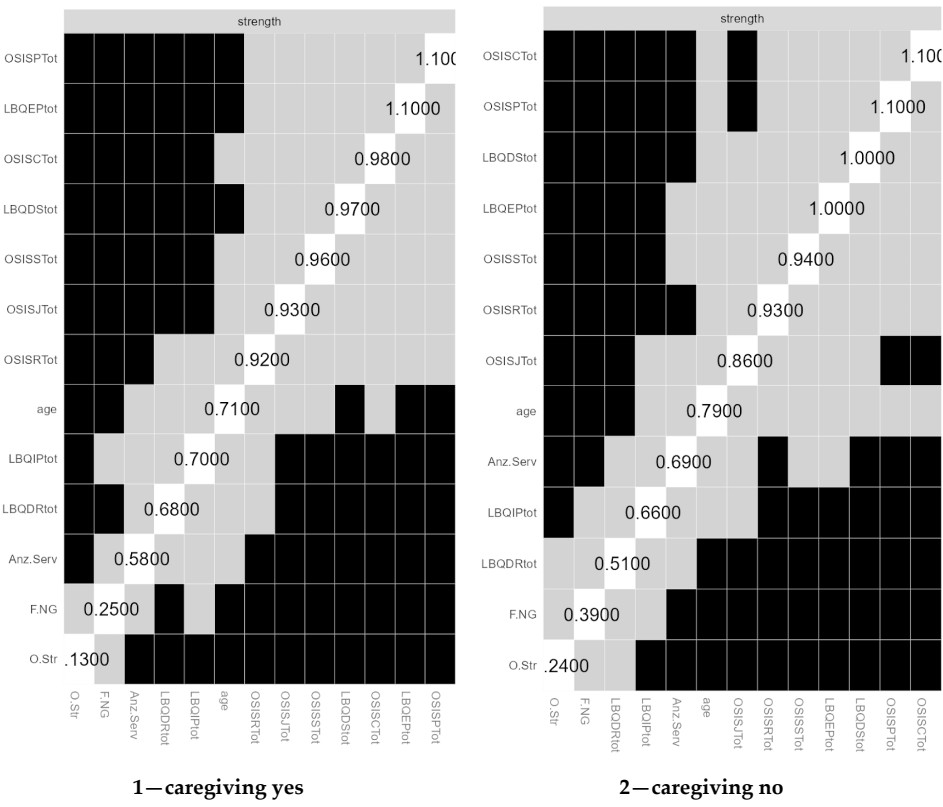

**1—caregiving yes**  **2—caregiving no**

**Figure A2.** Centrality stability coefficients per variable in the network regarding two groups (1—caregiving yes/2—caregiving no), obtained by non-parametric bootstrapped difference test for strength (gray squares designate no difference between nodes; black squares identify significant difference for $\alpha = 0.05$; in the diagonal, we observe the strength values of each node). Anz.Serv = seniority of service; O.Str = accumulated overtime hours; F.NG = unused vacation days; LBQEPtot = psychophysical exhaustion–engagement; LBQDRtot = relational deterioration–involvement; LBQIPtot = professional inefficacy–efficacy; LBQDStot = disillusion–fulfilment; OSISCtot = career satisfaction; OSISJtot = satisfaction with the job itself; OSISStot = satisfaction with the setting and the organizational structure; OSISPtot = satisfaction with organizational processes; OSISRtot = satisfaction with interpersonal relationships.

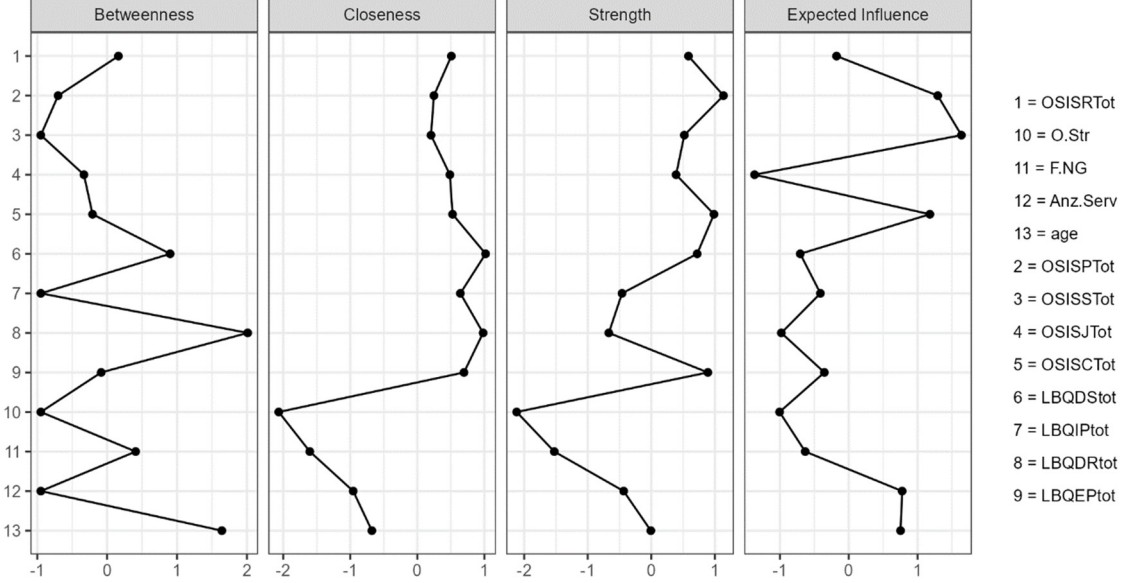

**Figure A3.** Centrality plot in total sample.

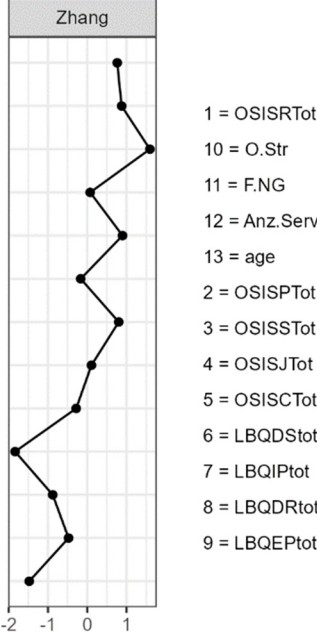

**Figure A4.** Clustering plot in total sample. Anz.Serv = seniority of service; O.Str = accumulated overtime hours; F.NG = unused vacation days; LBQEPtot = psychophysical exhaustion–engagement; LBQDRtot = relational deterioration–involvement; LBQIPtot = professional inefficacy–efficacy; LBQD-Stot = disillusion–fulfilment; OSISCtot = career satisfaction; OSISJtot = satisfaction with the job itself; OSISStot = satisfaction with the setting and the organizational structure; OSISPtot = satisfaction with organizational processes; OSISRtot = satisfaction with interpersonal relationships.

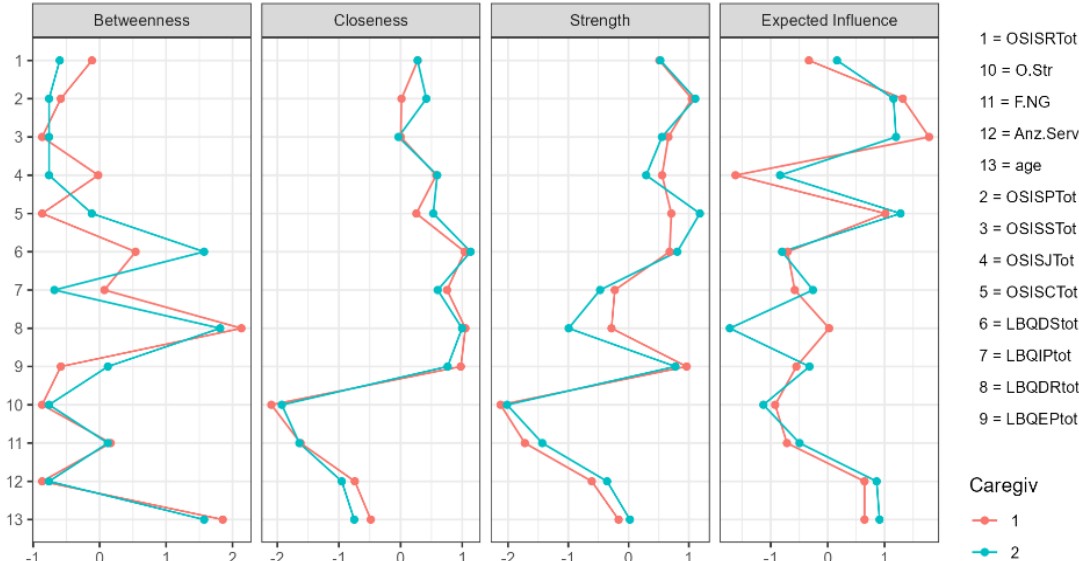

**Figure A5.** Centrality plot comparing two groups (caregiving yes/no). Anz.Serv = seniority of service; O.Str = accumulated overtime hours; F.NG = unused vacation days; LBQEPtot = psychophysical exhaustion–engagement; LBQDRtot = relational deterioration–involvement; LBQIPtot = professional inefficacy–efficacy; LBQDStot = disillusion–fulfilment; OSISCtot = career satisfaction; OSISJtot = satisfaction with the job itself; OSISStot = satisfaction with the setting and the organizational structure; OSISPtot = satisfaction with organizational processes; OSISRtot = satisfaction with interpersonal relationships.

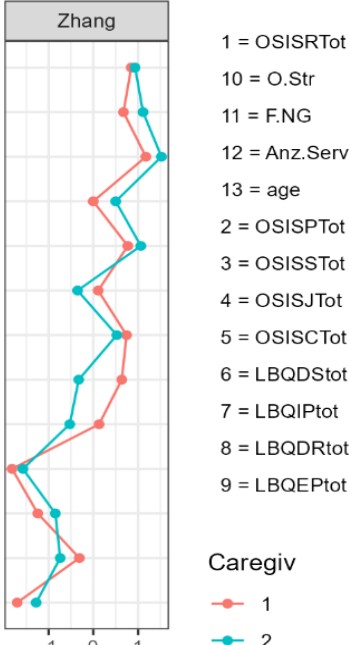

**Figure A6.** Clustering plot comparing two groups (caregiving yes/no). Anz.Serv = seniority of service; O.Str = accumulated overtime hours; F.NG = unused vacation days; LBQEPtot = psychophysical exhaustion–engagement; LBQDRtot = relational deterioration–involvement; LBQIPtot = professional inefficacy–efficacy; LBQDStot = disillusion–fulfilment; OSISCtot = career satisfaction; OSISJtot = satisfaction with the job itself; OSISStot = satisfaction with the setting and the organizational structure; OSISPtot = satisfaction with organizational processes; OSISRtot = satisfaction with interpersonal relationships.

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
