# Peer review of "Double-Duty Caregiving, Burnout, Job Satisfaction, and the Sustainability of the Work–Life Balance Among Italian Healthcare Workers: A Descriptive Study"

_sustainability, doi:10.3390/su17010039_

Round 1
Reviewer 1 Report
Comments and Suggestions for Authors
The article on the stress-satisfaction relation of double caregivers is build on a very important issue. The authors present a good review of theoretical reasons for the importance of the topic. Yet they fail in presenting sepcific hypotheses bases on the literature review. Specifically, (Issue 1) in the paragraph 1.4 (or around it) i would expect sepcific hypothes, bases on the literature review. They should be testable with the methods uses.
This has consequences to the analyitical part of the article. There are smaller or bigger issues there, which i will list below
1. Paragraph 2.1 - Readers would need to know what was specifically the " non-probabilistic sampling procedure" of research participants. Anybody willing? Friends and collegues? Those who needed extra cash for study participatinons? Volunteers responding to the styudy advertisments?
The recuritment method can have huge consequenes on the results.
2. The KEY independent variable is the performance of double duty caregiving. Yet we know very little of how it was assesed (besides the nominal form YES/NO to the question if a person feels like it). In my opinion this is also a huge disadvatage of the study, as a person who for examples ONLY takes kids to school each morning, would be signifficantly less strained than the person who substitues for an in-home nurse taking care of a disabled parrent. Yet due to the scale used they would be both be counted as beeing doube duty. This issue need to be cleared, and at best some distinctions here are needed (which might signifficantly influence the results).
3. Data Analysis - at the start of the paragraph 2.4 authors indicate they reviewed the data, yet they do not give any proof of that. Afterwards they list the used statistics, namely Pearson correlation and t-test, which are apropriate only for normaly distributed variables. We do not know if this is the case. Additionally, in line 211 authors indicate using Pearson's t-test to explore the data. It is NOT the purpose of this test!
4. I appreciate the usage of Network Analysis. It is one of the key distinguishing features of the article and should be kept if the authors decide to prepare a revised version. Yet the selection of variables for the analysis makes all the difference. First, the authors write about selecting SOME demographic variables (line 220). Why those (selected) and not others? The reason apearing in line 228 is not fully convincing, especially without giving the parameters for those variables (like normality, skeweness, curtosis, etc).
Secondly, the selection of variables without a strong theoretical backing will give results which are overfitted to the data. As an example, the strong (and obvious) correlation between age and tennure (seniority of service) "uses up" the available variance, thus artificially LOWERING other correlations in the NA. The same goes for the relation between satisfaction and burnout (which for many scholars are the two sides of the same coin!)
To colclude, the article need FIRST clearly stated hypotheses, which then would be tested/verified. This is not the case in the presented paper.
Author Response
Kind Reviewer#1, please see the attached Point by Point Letter

Reviewer 2 Report
Comments and Suggestions for Authors
1. Clearly articulate the study’s objectives by explicitly stating the research questions and hypotheses.
2. Enhance the contextual background with recent references that address contemporary challenges, particularly the effects of post-COVID-19 dynamics on healthcare workers.
3. Emphasize the study’s unique contribution to the existing literature to bolster its originality and relevance.
4. Justify the selection of instruments, such as the Link Burnout Questionnaire (LBQ) and the Occupational Stress Indicator (OSI), by detailing their significance in relation to the study’s objectives.
5. Provide additional information on how the variables were selected and how they align with the research focus.
6. Ensure transparency in the methodological approach, particularly regarding the inclusion and exclusion criteria for participants and data sources.
7. Present a more comprehensive narrative to accompany tables and figures, particularly network analysis visualizations, to aid readers in understanding key findings and trends.
8. Highlight significant patterns, correlations, and unexpected results, linking them to the study's objectives and broader implications.
9. Expand the critical analysis by comparing the findings with existing literature and addressing any conflicting results.
10. Discuss the broader implications of the findings for occupational health and their relevance to practitioners and policymakers.
11. Offer suggestions for future research, such as conducting longitudinal studies or exploring additional variables that may influence burnout and job satisfaction.
12. Provide actionable recommendations for healthcare organizations and policymakers, focusing on strategies to reduce burnout and enhance job satisfaction.
13. Propose specific interventions, such as tailored wellness programs, workload adjustments, or enhanced support systems for caregivers managing multiple responsibilities.
Comments on the Quality of English Language
Address minor grammatical issues and improve the sentence flow for better readability.
Conduct a thorough proofreading to enhance the overall quality of the manuscript.
Author Response
Kind Reviewer#2, please see the attached Point by Point Letter

Round 2
Reviewer 1 Report
Comments and Suggestions for Authors
Thanks for considering and implementinh my comments. There are still some points which could be written differently, but I belive the improvementsare sufficient for the article to be published. Merry Christmas!